# PAAN: Pyramid Attention Augmented Network for polyp segmentation

**Sida Yi**                                                                                YISIDA@STU.PKU.EDU.CN
**Jianhao Xie**                                                                  2301212751@STU.PKU.EDU.CN
**Hao-Cheng Kan**                                                              KANHORST@STU.PKU.EDU.CN
**Yuesheng Zhu**                                                                              ZHUYS@PKU.EDU.CN
**Guibo Luo**                                                                                    LUOGB@PKU.EDU.CN
*Peking University, School of Electronic and Computer Engineering, Shenzhen Graduate School*

**Editors:** Accepted for publication at MIDL 2024

## Abstract

Polyp segmentation is a task of segmenting polyp lesion regions from normal tissues in medical images, which is crucial for medical diagnosis and treatment planning. However, existing methods still suffer from low accuracy in polyp boundary delineation and insufficient suppression of irrelevant background due to the blurred boundaries and textures of polyps. To overcome these limitations, in this paper a Pyramid Attention Augmented Network (PAAN) is proposed, in which a pyramid feature diversion structure with spatial attention mechanism is developed so that good feature representation with low information loss can be achieved by conducting channel attention-based feature diversion and inter-layer fusion, while reducing computational complexity. Also, our framework includes an Enhanced Spatial Attention module (ESA), which can improve the quality of initial polyp segmentation predictions through spatial self-attention mechanism and multi-scale feature fusion. Our approach is evaluated on five challenging polyp datasets— Kvasir, CVC-ClinicDB, CVC-300, ETIS, and CVC-colonDB and achieves excellent results. In particular, we achieve 94.2% Dice and 89.7% IoU on Kvasir, outperforming other state-of-the-art methods.
**Keywords:** medical image segmentation, polyp segmentation, spatial attention

## 1. Introduction

Polyp segmentation is a crucial task in medical image analysis, as it plays a vital role in the early detection and diagnosis of colorectal cancer (Hsu et al., 2021). Colorectal cancer is the third most common type of cancer worldwide, accounting for 9.4% of all cancer-related deaths (Liu et al., 2021). Colonoscopy is an effective technique for colorectal cancer screening and prevention. The accurate and effective segmentation of polyp regions from endoscopic images directly influences medical decisions and patient treatment outcomes. However, polyps acquired from colonoscopy images are subject to background interference such as specular reflections and tissue occlusions, while the polyps exhibit variable shapes and textures, along with unclear boundaries with normal tissues (Sun et al., 2023). Therefore, achieving accurate polyp segmentation from medical images is a challenging task.

Early polyp segmentation methods rely on manually annotated features (Mamonov et al., 2014), such as color, texture, and shape combinations. These methods have high false positive rates due to the limited representation of manually extracted features. The emergence of deep learning provides promising approaches for accurate polyp segmentation.

U-Net (Ronneberger et al., 2015) captures complex patterns and restores detailed features through a symmetrical encoder-decoder U-shaped structure. Building upon this, PraNet (Fan et al., 2020) generates initial predictions using parallel decoders and applies reverse attention to generate more precise polyp boundaries. TGANet (Tomar et al., 2022) improves feature representation by introducing text-guided attention. Poly-SAM (Li et al., 2023) is applied to the polyp segmentation by fine-tuning the SAM (Kirillov et al., 2023).

Although the above methods have made certain progress in polyp segmentation, they still have the following limitations: insufficient extraction of features from different hierarchical levels and fusion of interlayer information in the encoder stage, which impacts the segmentation accuracy of blurry boundaries. In the initial decoder stage, there is inadequate attention to the spatial relationship and multi-scale features of polyps, resulting in insufficient suppression of irrelevant background in the initial predicted map and affecting the quality of subsequent decoder generation. Insufficient attention to uncertain regions leads to inaccurate boundary delineation.

To address identified challenges, we present the Pyramid Attention Augmented Network (PAAN), aimed at precisely delineating polyp boundaries and reducing background noise. PAAN features a pyramid feature diversion structure with channel attention to enhance feature extraction across different layers, ensuring minimal information loss and improved feature significance. An Enhanced Spatial Attention (ESA) module is introduced early in the decoding process to refine segmentation accuracy through spatial attention and multi-scale information fusion. In addition, we refer to and improve the framework of UACANet (Kim et al., 2021) to identify and learn from uncertain regions, aligning predictions more closely with ground truth. Our contributions are as follows:

- We propose the Pyramid Feature Diversion module (PFD), which effectively captures important features and reduces information loss by sequentially splitting and merging the channel attention at different levels while reducing computational complexity.
- We introduce the Enhanced Spatial Attention (ESA) in the decoder, enhancing the suppression of irrelevant background while improving multi-scale feature extraction.
- Extensive experiments on multiple datasets demonstrate that our method outperforms existing methods both qualitatively and quantitatively.

Due to space limitations, more related work is relegated to the Appendix, which can be found in Appendix B.

## 2. Methodology

Learning more powerful feature representations can improve the discriminative ability of models (Su et al., 2022). In this paper, we propose the Pyramid Attention Augmentation Network (PAAN), which is an enhanced model based on channel attention and spatial attention mechanisms. The overall framework of PAAN is shown in Figure 1. The loss function is described in Appendix A.1.

### 2.1. Overview of PAAN

PAAN adopts an encoder-decoder structure, where the encoder captures the context and the decoder restores detailed features. The parallel PFD-e encoder aggregates features to

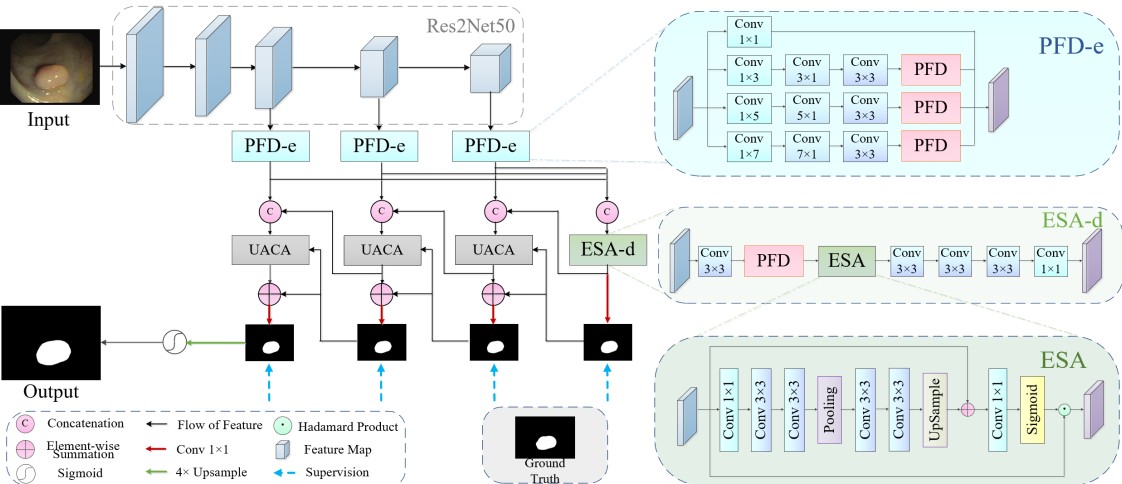

Figure 1: The architecture of the PAAN consists of PFD-e, a multi-scale pyramid feature diversion encoder, and ESA-d, a spatial attention decoder.

generate the initial prediction map via ESA-d, while the decoder receives information from the encoder and the previous decoder layer to output accurate prediction maps step by step.

The original image is first input into the Res2Net50 (Gao et al., 2019) backbone for feature extraction. By gradually reducing the size of the feature maps and increasing the number of channels, deep feature maps contain more abstract semantic information compared to shallow ones (Wu et al., 2019), while also reducing computation. Therefore, we only consider the deep feature maps $f_3$, $f_4$, and $f_5$ at layers $i$=3, 4, and 5 as inputs. Each level of the input feature maps has a size of $\{\frac{h}{2^{i-1}}, \frac{w}{2^{i-1}}\}$. The PFD-e module serves two purposes: 1. Concatenating its output with the input of the next decoder module at a deeper level. 2. Allowing the three different levels of PFD modules to collectively input the decoder ESA-d to generate the initial prediction map.

In the PFD-e module, the feature maps from the backbone are input into the Pyramid Feature diversion module (PFD in Figure 2) through pathways with different receptive fields (1×3,5,7) and then concatenated as output. The PFD consists of two key components: the Feature Diversion Attention module (FDA) and the Hierarchical Interactive Attention module (HIA). In the FDA, only half of the channels are fed into the next FDA module, while the other half is directly fused at the HIA, which helps reduce computational complexity and improve attention effectiveness in the network. In the decoder ESA-d that generates the initial prediction map, we introduce the Enhanced Spatial Attention module (ESA), which can adaptively generate masks based on the input features, thereby improving the accuracy of polyp segmentation. The initial prediction map is progressively enhanced through several UACA modules and is processed with the Sigmoid function and 4x bilinear upsampling to generate the final segmentation map. UACA is the Uncertainty Augmented Decoder in UACANet (Kim et al., 2021), which can work well with the proposed module.

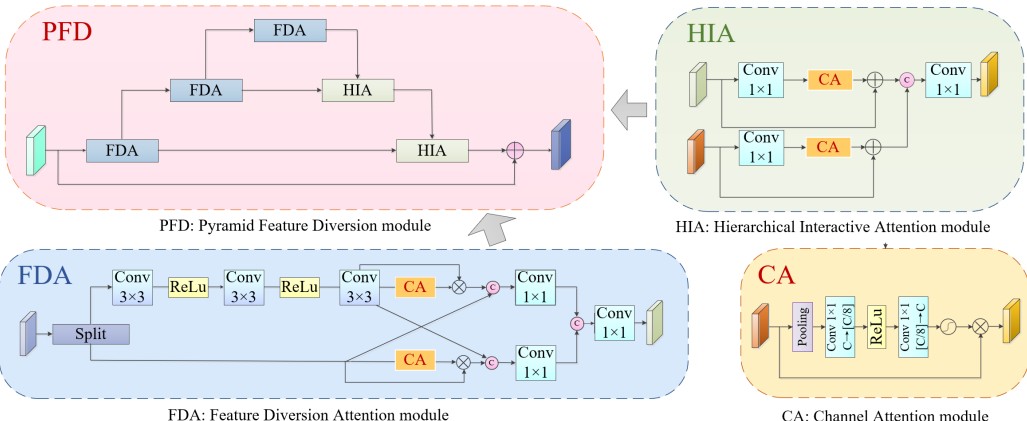

Figure 2: The FDA and HIA module in the PFD.

## 2.2. Pyramid Feature Diversion encoder (PFD-e)

In PFD-e, multi-scale polyp features are captured through the convolutional receptive fields of different kernel sizes. Deep feature maps from the backbone at levels $i$=3, 4, 5 enter four pathways with distinct receptive fields. The pathway with receptive fields (1×3,5,7) further extracts features through the PFD. The features undergo three levels of distribution: (1) different depths in the backbone, (2) different receptive fields in PFD-e, (3) different levels in the PFD. This hierarchical feature extraction process ensures precise separation and extraction of different features at each stage of the decoder, thereby creating a strong feature representation for polyp segmentation.

### 2.2.1. Pyramid Feature Diversion module (PFD)

Multilevel feature fusion is crucial in dense prediction tasks (Su et al., 2022) as low-level and high-level features are complementary. In the pyramid structure of the PFD shown in Figure 2, the feature extractor separates features into different abstraction levels in the upward phase and gradually fuses them in the downward phase, providing a method to extract higher-level feature representations.

In the PFD, the feature flows through a feature distribution and fusion network with a pyramid shape. The pyramid consists of 3 FDA modules and 2 HIA modules. Both modules utilize channel attention mechanisms (Hu et al., 2018) (Implemented by the CA module) to achieve adaptive channel weights and feature selection. Each FDA module takes the output of the lower-level FDA as input and enhances feature selection through cascaded connections. Each HIA fuses the output from this layer and the high-level FDA and achieves feature compression and extraction through feature fusion. The PFD achieves the purpose of better extraction of feature representation while reducing computation.

**(1) Feature Diversion Attention module (FDA):** In the PFD, the FDA is used to divide the feature map $m$ into two branches $x_1$ and $x_2$ with equal channel numbers and apply different processing to the two branches. In the first stage, $x_1$ is subjected to a series of convolutions and ReLU activation for feature extraction, while no operations are performed on $x_2$. In the second stage, $x_1$ and $x_2$ are input into the CA module to obtain

feature maps $x_2 \odot CA(x_1)$ and $x_2 \odot CA(x_2)$ with adaptive channel weights, which are then concatenated with the original inputs $x_1$ and $x_2$. The feature map after concatenation is finally fused and compressed by a $1 \times 1$ convolution layer to obtain the final output.

In summary, the FDA module adaptively adjusts the importance of different channels in the feature map through channel feature diversion and channel attention mechanisms, thereby generating more informative and expressive output feature maps.

**(2) Hierarchical Interactive Attention module (HIA):** The HIA module aims to fuse the FDA outputs $x_1$ and $x_2$ from the current layer and the previous layer, and enhance the channel attention and compress the channels. Similar to the FDA module, $x_1$ and $x_2$ are fed into the CA module for channel attention enhancement, and the outputs $x_2 \odot CA(x_1)$ and $x_2 \odot CA(x_2)$ are obtained. Then, they are concatenated and compressed to obtain the output. In summary, the HIA module performs inter-layer fusion using the channel attention mechanism, focusing on the feature representations of different layers, and generating more informative and expressive output feature maps.

## 2.3. Enhanced Spatial Attention decoder (ESA-d)

The decoder ESA-d is designed to generate initial saliency maps for prediction, which is crucial for subsequent decoding processes. The ESA-d receives output feature maps $f_1$, $f_2$, and $f_3$ from different levels of the backbone network PAA. To match the feature map sizes, $f_1$ and $f_2$ are upsampled by 4 times and 2 times respectively, and $f_3$ is the original size feature map. These feature maps are concatenated to generate the input feature map $f_{in}$. The feature shuffling structure is applied to $f_{in}$ to separate the feature representations of different levels after concatenation, and then input into the ESA module.

### 2.3.1. ENHANCED SPATIAL ATTENTION MODULE (ESA)

The ESA module aims to generate a feature mask to enhance the important spatial regions of the input feature maps. The module first applies a 1x1 convolutional layer to compress the number of channels to 1/4, and further extracts and compresses spatial information through downsampling. The maximum pooling operation selects the maximum value in each region as the salient feature, providing the basis for generating spatial attention maps. Meanwhile, a series of convolutional layers are applied for feature processing, which can extract features at different spatial scales and help understand different parts of the scene. Finally, the feature map size is adjusted back to the size of the input feature map using bilinear interpolation, and the adjusted feature map is fused with the original map, resulting in pixel-wise enhancement. The spatial mask is generated by Sigmoid, effectively combining deep spatial features and shallow spatial features. The specific formula is as follows:

$$f' = \mathcal{B}(f_i), f_o = f_i \odot \sigma(cls(f_i + f')) \tag{1}$$

where $f_i$ and $f_o$ are the input and output of ESA, $\mathcal{B}$ represents the spatial feature extraction which includes pooling and upsampling, and $\sigma$ and $cls$ denote Sigmoid and 1×1 convolution.

In summary, the ESA enhances the network's attention ability to specific regions through the self-attention mechanism and feature transformation operations, extracts more accurate and region-specific feature representations, and generates better output feature maps.

## 3. Experiments

In this section, we present the experimental setup and results. The results obtained by PAAN are compared quantitatively and qualitatively with six state-of-the-art polyp segmentation methods. Appendix A.2-A.4 provides details about the experimental setup, datasets, and evaluation metrics. **The ablation study is described in Appendix C.**

### 3.1. Experimental Setup, Datasets, and Metrics

PAAN is trained on NVIDIA TITAN 3090 GPU in Pytorch. We used Adam for optimization, with a learning rate of 0.0001, a batch size of 16, and 1000 epochs. We use the following 5 datasets for experiments and evaluations: CVC-ClinicDB (Bernal et al., 2015), CVC-ColonDB (Tajbakhsh et al., 2015), ETIS (Silva et al., 2014), Kvasir (Jha et al., 2020), and CVC-300 (Vázquez et al., 2017). 1450 of these images were used for training and 800 for testing. The evaluation metrics used in this paper are IoU, Dice, MAE, $F_\beta^w$, $S_\alpha$, and $E_\phi^{mean}$. These metrics provide multifaceted insight into both accuracy and robustness. More details are in Appendix A.2-A.4.

### 3.2. Experimental Results

The quantitative results of PAAN on five polyp datasets compared with the state-of-the-art methods are presented in Table 1. PAAN demonstrates outstanding performance in terms of IoU, Dice, MAE, and surpasses the existing methods on all datasets. In particular, compared to the previous state-of-the-art model (i.e., UACANet-L), our average Dice scores on Kvasir and EITS have improved by 3.0% and 4.7% respectively. Furthermore, our network has also achieved significant improvements on the other four metrics: MAE, $F_\beta^w$, $S_\alpha$, and $E_\phi^{mean}$.

Figure 3 presents the visual results of our network and contrast methods on multiple datasets. We intentionally selected challenging polyp images with smaller and more blurred regions compared to typical polyp images. Additionally, there is significant interference from lighting and irrelevant features. The results show that 6 comparative methods exhibit inaccurate judgments of blurry boundaries or mistakenly identify reflective irrelevant areas as polyp regions. In the 3rd row, other methods are affected by background interference, resulting in the mislabeling of irrelevant regions. In the 4th row, the under-segmentation occurs due to the small size of the polyp area. In contrast, the PAAN can accurately identify polyps, effectively capturing their boundaries and texture changes, while excluding interference from irrelevant background information. This highlights the advantages of our attention-enhanced network in detail recognition and background suppression. By utilizing visual methods, researchers can have a clearer observation and understanding of the segmentation results of PAAN, as well as its relative advantages in perceiving fine details, handling fuzzy boundaries, and suppressing irrelevant information.

### 3.3. Discussion

By comparing the qualitative and quantitative analysis results of other networks, it can be seen that the method proposed in this paper has the following advantages:

(1) **Accuracy of results:** In terms of quantitative results, it can be seen that the networks in this paper outperform competitors in major indicators such as mDice, mIoU, and

Table 1: Comparison to the previous state-of-the-art methods on 5 polyp datasets.

| Dataset | Method | mDice ↑ | mIoU ↑ | MAE ↓ | $F_\beta^w$ ↑ | $S_\alpha$ ↑ | $E_\phi^{mean}$↑ |
|---|---|---|---|---|---|---|---|
| Kvasir | U-Net (Ronneberger et al., 2015) | 0.818 | 0.746 | 0.055 | 0.794 | 0.858 | 0.893 |
| | U-Net++ (Zhou et al., 2018) | 0.821 | 0.743 | 0.048 | 0.808 | 0.862 | 0.910 |
| | ResUNet++ (Jha et al., 2019) | 0.807 | 0.727 | 0.052 | 0.777 | 0.840 | 0.882 |
| | PraNet (Fan et al., 2020) | 0.898 | 0.840 | 0.030 | 0.885 | 0.915 | 0.948 |
| | UACANet-S (Kim et al., 2021) | 0.905 | 0.852 | 0.026 | 0.897 | 0.914 | 0.948 |
| | UACANet-L (Kim et al., 2021) | 0.912 | 0.859 | 0.025 | 0.902 | 0.917 | 0.955 |
| | CaraNet (Lou et al., 2022) | 0.918 | 0.865 | 0.023 | 0.909 | 0.929 | 0.968 |
| | FAPN (Su et al., 2022) | 0.913 | 0.865 | 0.023 | 0.908 | 0.922 | 0.952 |
| | **Ours** | **0.942** | **0.897** | **0.015** | **0.936** | **0.942** | **0.977** |
| CVC-ClinicDB | U-Net (Ronneberger et al., 2015) | 0.823 | 0.755 | 0.019 | 0.811 | 0.889 | 0.954 |
| | U-Net++ (Zhou et al., 2018) | 0.794 | 0.729 | 0.022 | 0.785 | 0.873 | 0.931 |
| | ResUNet++ (Jha et al., 2019) | 0.846 | 0.786 | 0.014 | 0.840 | 0.891 | 0.939 |
| | PraNet (Fan et al., 2020) | 0.899 | 0.849 | 0.009 | 0.896 | 0.936 | 0.979 |
| | UACANet-S (Kim et al., 2021) | 0.916 | 0.870 | 0.009 | 0.917 | 0.940 | 0.965 |
| | UACANet-L (Kim et al., 2021) | 0.926 | 0.880 | 0.006 | 0.928 | 0.943 | 0.974 |
| | FAPN (Su et al., 2022) | 0.931 | 0.879 | 0.008 | 0.929 | 0.941 | 0.983 |
| | **Ours** | **0.934** | **0.884** | **0.007** | **0.932** | **0.946** | **0.985** |
| ETIS | U-Net (Ronneberger et al., 2015) | 0.398 | 0.335 | 0.036 | 0.366 | 0.684 | 0.740 |
| | U-Net++ (Zhou et al., 2018) | 0.401 | 0.344 | 0.035 | 0.390 | 0.683 | 0.776 |
| | ResUNet++ (Jha et al., 2019) | 0.337 | 0.271 | 0.044 | 0.313 | 0.622 | 0.636 |
| | PraNet (Fan et al., 2020) | 0.628 | 0.567 | 0.031 | 0.600 | 0.794 | 0.841 |
| | UACANet-S (Kim et al., 2021) | 0.694 | 0.615 | 0.023 | 0.650 | 0.815 | 0.848 |
| | UACANet-L (Kim et al., 2021) | 0.766 | 0.689 | 0.012 | 0.740 | 0.859 | 0.903 |
| | CaraNet (Lou et al., 2022) | 0.747 | 0.672 | 0.017 | 0.709 | 0.868 | 0.894 |
| | FAPN (Su et al., 2022) | 0.780 | 0.715 | 0.012 | 0.757 | 0.870 | 0.910 |
| | **Ours** | **0.813** | **0.734** | **0.010** | **0.789** | **0.882** | **0.938** |
| CVC-ColonDB | U-Net (Ronneberger et al., 2015) | 0.512 | 0.444 | 0.061 | 0.498 | 0.712 | 0.776 |
| | U-Net++ (Zhou et al., 2018) | 0.483 | 0.410 | 0.064 | 0.467 | 0.691 | 0.760 |
| | ResUNet++ (Jha et al., 2019) | 0.588 | 0.497 | 0.058 | 0.551 | 0.729 | 0.772 |
| | PraNet (Fan et al., 2020) | 0.709 | 0.640 | 0.045 | 0.696 | 0.819 | 0.869 |
| | UACANet-S (Kim et al., 2021) | 0.783 | 0.704 | 0.034 | 0.772 | 0.848 | 0.894 |
| | UACANet-L (Kim et al., 2021) | 0.751 | 0.678 | 0.039 | 0.746 | 0.835 | 0.875 |
| | CaraNet (Lou et al., 2022) | 0.773 | 0.689 | 0.042 | 0.729 | 0.853 | 0.902 |
| | FAPN (Su et al., 2022) | 0.785 | 0.706 | 0.033 | 0.777 | 0.849 | 0.890 |
| | **Ours** | **0.786** | **0.716** | **0.033** | **0.783** | **0.854** | **0.894** |
| CVC-300 | U-Net (Ronneberger et al., 2015) | 0.710 | 0.627 | 0.022 | 0.684 | 0.843 | 0.876 |
| | U-Net++ (Zhou et al., 2018) | 0.707 | 0.624 | 0.018 | 0.687 | 0.839 | 0.898 |
| | ResUNet++ (Jha et al., 2019) | 0.687 | 0.598 | 0.022 | 0.650 | 0.811 | 0.816 |
| | PraNet (Fan et al., 2020) | 0.871 | 0.797 | 0.010 | 0.843 | 0.925 | 0.972 |
| | UACANet-S (Kim et al., 2021) | 0.902 | 0.837 | 0.006 | 0.886 | 0.934 | 0.974 |
| | UACANet-L (Kim et al., 2021) | 0.910 | 0.849 | 0.005 | 0.901 | 0.937 | 0.977 |
| | CaraNet (Lou et al., 2022) | 0.903 | 0.838 | 0.007 | 0.887 | 0.940 | 0.989 |
| | FAPN (Su et al., 2022) | 0.910 | 0.847 | 0.005 | 0.896 | 0.939 | 0.975 |
| | **Ours** | **0.926** | **0.869** | **0.004** | **0.921** | **0.948** | **0.990** |

Table 2: Comparison of computational complexity and parameter count.

| Model | UACANet-S | UACANet-L | CaraNet | FAPN | Ours |
|---|---|---|---|---|---|
| Macs(G) | 12.04 | 59.57 | 21.7 | 27.02 | 20.94 |
| Params(M) | 26.9 | 69.16 | 46.64 | 35.61 | 33.72 |

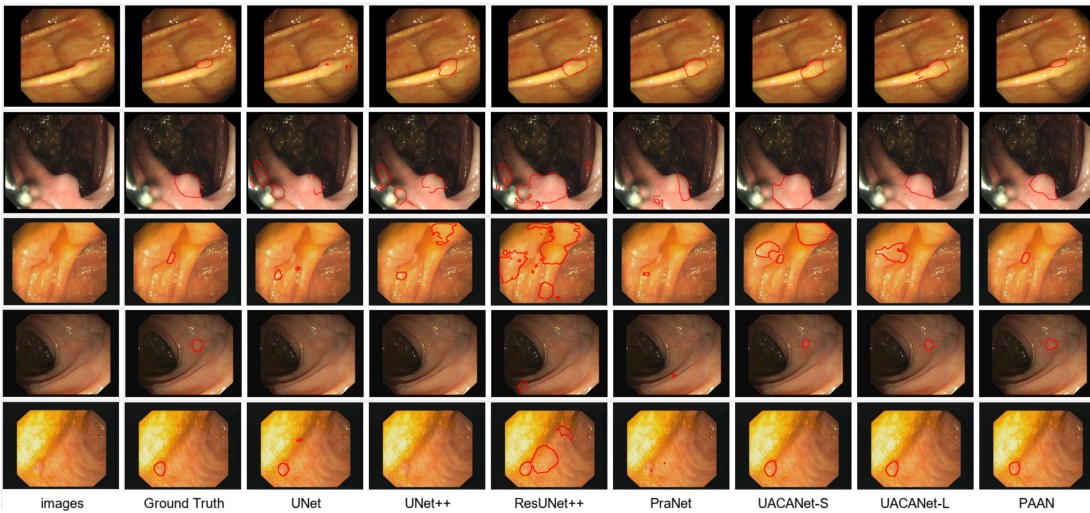

Figure 3: Qualitative results comparison with previous state-of-the-art methods.

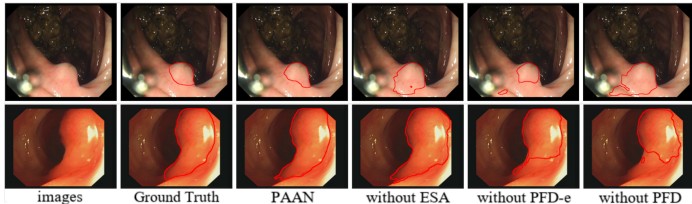

Figure 4: Visualization of ablation results. More results can be found in the Appendix F.

MAE. In terms of qualitative results, even in cases with fuzzy polyp boundaries and smaller polyp sizes, the networks in this paper can still accurately depict the polyp boundaries, which demonstrates the good resolution ability of pyramid feature diversion and spatial attention to fuzzy boundary details. The uncertainty analysis can be found in Appendix E.

(2) **Ability to suppress background interference:** By the qualitative results, it can be observed that our method has accuracy in capturing polyp boundaries and the ability to suppress background interference. Even in complex polyp scenes with lots of fuzzy features, PAAN can still accurately extract effective features and suppress irrelevant features. The ESA module improves the accuracy of segmentation results by better capturing the spatial dependencies in the input image. By enhancing spatial attention, the focus on polyp and the irrelevant background suppression are improved, thus improving the accuracy.

(3) **Lower computational complexity:** Our network improves segmentation with less complexity than previous methods. Detailed can be found in Table 2 and Appendix D.

## 4. Ablation Study

We conducted complete ablation experiments under all five datasets The quantitative experiments employed mDice, mIoU, and MAE as evaluation metrics, with the results shown in Table 3 and Figure 4. Ablation results prove the effectiveness of each module in this paper, and the complete PAAN network has better segmentation results under all datasets. Detailed discussion is provided in Appendix C.

Table 3: The ablation results of PFD-e, PFD, and ESA.

|  | Kvasir | | | CVC-ClinicDB | | | ETIS | | | CVC-ColonDB | | | CVC-300 | | |
|---|---|---|---|---|---|---|---|---|---|---|---|---|---|---|---|
|  | mDice | mIoU | MAE | mDice | mIoU | MAE | mDice | mIoU | MAE | mDice | mIoU | MAE | mDice | mIoU | MAE |
| w/o PFD-e | 0.858 | 0.785 | 0.036 | 0.786 | 0.697 | 0.029 | 0.697 | 0.591 | 0.016 | 0.669 | 0.579 | 0.043 | 0.869 | 0.784 | 0.008 |
| w/o PFD | 0.873 | 0.808 | 0.035 | 0.851 | 0.779 | 0.022 | 0.725 | 0.644 | 0.039 | 0.722 | 0.644 | 0.039 | 0.898 | 0.831 | 0.006 |
| w/o ESA | 0.895 | 0.873 | 0.023 | 0.816 | 0.746 | 0.022 | 0.743 | 0.655 | 0.012 | 0.719 | 0.646 | 0.039 | 0.893 | 0.825 | 0.006 |
| PAAN | 0.942 | 0.897 | 0.015 | 0.934 | 0.884 | 0.007 | 0.813 | 0.734 | 0.010 | 0.786 | 0.716 | 0.033 | 0.926 | 0.869 | 0.004 |

## 5. Conclusion

In this paper, we propose the Pyramid Attention Augmented Network (PAAN) for accurate polyp segmentation. Our network combines spatial attention mechanism and pyramid feature diversion structure to effectively capture important features and reduce information loss. The FDA and HIA module in the pyramid feature diversion structure reduce computational complexity and improve the effectiveness of attention. The Enhanced Spatial

Attention module utilizes spatial self-attention and multi-scale feature fusion to further improve the accuracy of polyp segmentation. Experiments on five challenging polyp datasets demonstrate that our network achieves excellent quantitative and qualitative results in terms of accuracy and robustness, outperforming existing methods.

## Acknowledgments

This work is supported by 2023 Shenzhen sustainable supporting funds for colleges and universities (No.20231121165240001), and Shenzhen Science and Technology Program (No. JCYJ20230807120800001).

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

## Appendix A. Experimental Detail

### A.1. Loss function

We use the binary cross-entropy loss function $\mathcal{L}_{BCE}$ and the intersection over union loss function $\mathcal{L}_{IoU}$. The loss functions are computed as follows:

$$
\begin{aligned}
\mathcal{L} &= \mathcal{L}_{BCE} + \mathcal{L}_{IoU}, \\
\mathcal{L}_{IoU} &= 1 - \frac{\sum_{i \in \mathcal{S}} y_i \hat{y}_i}{\sum_{i \in \mathcal{S}} y_i + \hat{y}_i - y_i \hat{y}_i}, \\
\mathcal{L}_{BCE} &= -\sum_{i \in \mathcal{S}} y_i log(\hat{y}_i) + (1 - y_i) log(1 - \hat{y}_i),
\end{aligned}
\tag{2}
$$

where $i$ denotes a single pixel in image $S$, $y_i$ and $\hat{y}_i$ represent the ground truth and the output, respectively. The binary cross-entropy (BCE) loss measures the difference between the output probability distribution and the true label distribution, aiming to make the model's output probability distribution as close as possible to the true label distribution. The intersection over union (IoU) loss measures the overlap between the predicted segmentation result and the true segmentation, to maximize the overlap between the predicted and the true segmentation regions.

### A.2. Experimental Setup

We implemented PAAN using the Python programming language in the PyTorch deep learning library, and trained and evaluated it on a high-performance computer equipped with an NVIDIA TITAN 3090 GPU. We used Adam for optimization, with a learning rate of 0.0001, a batch size of 16, and 1000 epochs. In addition, our network is capable of early stopping based on the validation set loss.

To evaluate the performance of PAAN, we used several evaluation metrics commonly used for polyp segmentation tasks, including IoU, Dice, MAE, $F_\beta^w$, $S_\alpha$, and $E_\phi^{mean}$. These metrics provide insight into the accuracy and robustness of our network's segmentation results.

### A.3. Datasets

In this paper, we use the following five datasets for experiments and evaluations: CVC-ClinicDB (Bernal et al., 2015), CVC-ColonDB (Tajbakhsh et al., 2015), ETIS (Silva et al., 2014), Kvasir (Jha et al., 2020), and CVC-300 (Vázquez et al., 2017). Below is a detailed introduction to each dataset and its characteristics:

- CVC-ClinicDB: This dataset was extracted from colonoscopies and contains 612 colon polyp images from 29 different sequences. The image resolution is 388×284. Each image is equipped with binary segmentation labels containing the correct outlines of colon polyps.

- CVC-ColonDB: Similar to CVC-ClinicDB, this dataset contains 380 colonoscopy images from 15 sequences. The image resolution is 574×500. Each image is also equipped with binary segmentation labels for accurate segmentation of colon polyps. Unlike CVC-ClinicDB, the images in CVC-ColonDB contain multiple colon polyps and other structures.

- ETIS: This dataset is a collection of colonoscopy images provided in the ETIS project. It contains 196 white-light images from 34 sequences with an image resolution of 1225×966. Each image has a corresponding binary label.

- Kvasir: This is a gastroscopy image dataset containing 1000 stomach images and corresponding binary segmentation labels, with various image resolutions and sizes. The Kvasir dataset is suitable for gastric polyp segmentation, with a wide variety of images covering polyps of different scales and shapes.

- CVC-300: CVC-300 is also a commonly used colonoscopy image dataset for colon polyp segmentation studies. The CVC-300 dataset contains 300 colonoscopy images, each containing colon polyps as well as other structures.

In order to make a fair comparison, we used the same data partition in this study. We used 1450 polyp images as training data, including 550 from CVC-ClinicDB and 900 from Kvasir. In the testing phase, we conducted tests on all 5 datasets, with a total of 800 images tested. Among them, CVC-ClinicDB consisted of 62 images, Kvasir consisted of 100 images, CVC-300 consisted of 62 images, CVC-ColonDB consisted of 380 images, and EITS consisted of 196 images.

### A.4. Evaluation metrics

In this section, we introduce six evaluation metrics used in this study, including Dice, IoU, MAE (Perazzi et al., 2012), $F_\beta$ (Borji et al., 2015), $S_\alpha$ (Fan et al., 2017), and $E_\phi$ (Fan et al., 2018).

(1) **Dice:** The Dice coefficient is a commonly used segmentation metric, which is defined as twice the intersection divided by the sum of the pixels, also known as the F1 score. It ranges from 0 to 1, where a value closer to 1 indicates a higher similarity between the prediction and the ground truth.

$$\text{Dice} = \frac{2 \times \text{TP}}{2 \times \text{TP} + \text{FP} + \text{FN}}. \tag{3}$$

where TP represents true positive, FP represents false positive, and FN represents false negative.

(2) **IoU:** Intersection-Over-Union, also known as the Jaccard index. IoU is the ratio of the overlap area between the predicted segmentation and the corresponding ground truth to the union area of the predicted segmentation and the ground truth (the intersection divided by the union). A value of 0 indicates no overlap, while a value of 1 represents a completely overlapping segmentation.

$$\text{IoU} = \frac{\text{TP}}{\text{TP} + \text{FP} + \text{FN}}. \tag{4}$$

(3) **MAE:** Mean Absolute Error (MAE) is a commonly used metric for evaluating the performance of a model. It measures the average absolute difference between the predicted values and the true values. A smaller MAE value indicates a higher prediction accuracy of the model. Compared to Mean Squared Error (MSE), MAE is more robust to outliers as it is not influenced by outliers. The formula to calculate MAE is as follows:

$$\text{MAE} = \frac{1}{W * H} \sum_{i=1}^{W} \sum_{i=1}^{H} |S_{i,j} - G_{i,j}|, \tag{5}$$

where $W$ and $H$ represent the width and height of the image, and $S_{i,j}$ and $G_{i,j}$ represent the corresponding pixels of the predicted image and the ground truth.

(4) $F_{\beta}$**:** Also known as F-measure or F1 score, it calculates the harmonic mean of precision and recall, with the weight adjusted by $\beta$. The calculation formula is as follows:

$$F_{\beta} = (1 + \beta^2) \frac{(\text{Precision} \cdot \text{Recall})}{(\beta^2 \cdot \text{Precision}) + \text{Recall}}. \tag{6}$$

(5) $S_{\alpha}$**:** The Structure-measure is used to evaluate the structural similarity of segmentation results.

$$S_{\alpha} = \alpha \cdot S_O + (1 - \alpha) \cdot S_R. \tag{7}$$

where $S_O$ represents object similarity, $S_R$ represents region similarity, and $\alpha$ is the weight adjustment parameter.

(6) $E_{\phi}$**:** Also known as Enhanced-measure, is used to evaluate the region coverage rate.

$$E_{\phi} = \frac{(1 + \phi^2) \cdot S_O \cdot S_R}{\phi^2 \cdot S_O + S_R}. \tag{8}$$

where $S_O$ represents object similarity, $S_R$ represents region similarity, and $\phi$ is a weight adjustment parameter.

## Appendix B. Related Works

### B.1. Medical Image Segmentation

Medical image segmentation is a specialized area that focuses on the analysis of anatomical structures or abnormalities in medical images. Due to its nature as a dense prediction task, the encoder-decoder architecture has been widely used. U-Net (Ronneberger et al., 2015) is one of the earliest methods in this field, which greatly promotes the application of deep learning in medical segmentation due to its high scalability and good performance with fewer annotations. Building upon this, UNet++ (Zhou et al., 2018) is proposed, which is an extension of the classic U-Net. By recursively constructing a deeper network through upsample and downsample pathways, it improves the segmentation performance. FSC-UNet (Chen et al., 2022), on the other hand, reduces the impact of feature maps from different abstraction levels on model parameters through submodule skip fusion. ScaleFormer (Huang et al., 2022) proposes a scale-oriented approach to improve the segmentation quality of medical images, demonstrating sensitivity towards small targets in multi-organ segmentation. DSCA-Net (Shan et al., 2022) through the combination of attention and depth-wise

separable convolution, enhances the recognition performance of electron microscopy images. These studies have demonstrated positive outcomes in various medical segmentation areas. However, specific feature extraction methods tailored to address challenges such as indistinct boundaries and diverse textures in polyp images are neglected in these methods.

### B.2. Polyp Segmentation

In the context of polyp segmentation, CANet (Zhang et al., 2019) demonstrates superior performance in colon polyp segmentation by the context aggregation method and cascaded self-attention module. PolypSeg (Zhong et al., 2020) effectively extracts features with rich semantic information by introducing the context-aware module and adaptive attention module. UACANet (Kim et al., 2021) captures polyp boundary information more effectively by introducing attention mechanism to uncertain regions during feature extraction and segmentation. CASCADE (Rahman and Marculescu, 2023) alleviates the problem of inconsistent feature size by introducing Transformer and attention-based convolutional modules. CaraNet (Lou et al., 2022) improves the segmentation performance of small medical objects by introducing a context-oriented directional reverse attention mechanism. Although these methods achieve better performance than manually crafted features, there is still room for improvement in polyp feature extraction, spatial representation, and irrelevant background suppression, and new feature extraction and spatial attention need to be introduced.

## Appendix C. Ablation Study

To validate the effectiveness of the mentioned modules in the paper, we conducted comprehensive qualitative and quantitative experiments on five datasets. The quantitative experiments employed mDice, mIoU, and MAE as evaluation metrics, with the results shown in Table 4, while the qualitative experimental results are presented in Figure 5.

**Ablation Study for PFD and PFD-e:** To validate the effectiveness of PFD, we trained and evaluated PAAN without the PFD module. One group of experiments completely removed PFD-e, while the other group of experiments retained the multi-scale framework of PFD-e while removing the PFD, specifically focusing on investigating the role of attention in the pyramid feature diversion structure of PFD. We conducted tests on all five test datasets, and the results are shown in Table 4. The performance gap between "w/o PFD" and "PAAN" is significantly larger than the gap between "w/o PFD" and "w/o PFD-e". This indicates that the effect of a simple multi-scale receptive field is not ideal, and the advantages of enhanced feature extraction and better feature representation brought by the pyramid attention structure are crucial. Additionally, as can be seen in Figure 5, the poor handling of blurry boundary features without PFD results in scattered boundary regions detached from the main subject, demonstrating the effectiveness of the feature fusion structure employed in this study.

**Ablation Study for ESA:** To validate the effectiveness of ESA, we removed the ESA from the ESA-d decoder while retaining the decoding module based on multiple convolutional operations. This allowed us to specifically investigate the role of spatial attention in the initial decoder. We conducted tests on the same evaluation metrics on five datasets, and the results are shown in Table 4. The ESA has shown improvements in metrics such as mDice, mIoU, and MAE. Furthermore, as seen in Figure 5, the lack of ESA leads to

Table 4: The ablation results of PFD-e, PFD, and ESA.

| | Kvasir | | | CVC-ClinicDB | | | ETIS | | | CVC-ColonDB | | | CVC-300 | | |
|---|---|---|---|---|---|---|---|---|---|---|---|---|---|---|---|
| | mDice | mIoU | MAE | mDice | mIoU | MAE | mDice | mIoU | MAE | mDice | mIoU | MAE | mDice | mIoU | MAE |
| w/o PFD-e | 0.858 | 0.785 | 0.036 | 0.786 | 0.697 | 0.029 | 0.697 | 0.591 | 0.016 | 0.669 | 0.579 | 0.043 | 0.869 | 0.784 | 0.008 |
| w/o PFD | 0.873 | 0.808 | 0.035 | 0.851 | 0.779 | 0.022 | 0.725 | 0.644 | 0.039 | 0.722 | 0.644 | 0.039 | 0.898 | 0.831 | 0.006 |
| w/o ESA | 0.895 | 0.873 | 0.023 | 0.816 | 0.746 | 0.022 | 0.743 | 0.655 | 0.012 | 0.719 | 0.646 | 0.039 | 0.893 | 0.825 | 0.006 |
| PAAN | 0.942 | 0.897 | 0.015 | 0.934 | 0.884 | 0.007 | 0.813 | 0.734 | 0.010 | 0.786 | 0.716 | 0.033 | 0.926 | 0.869 | 0.004 |

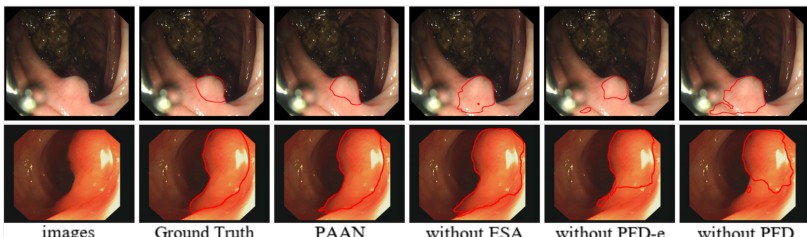

Figure 5: Visualization of ablation experimental results. More results can be found in the Appendix F

insufficient background suppression, resulting in an expansion bias in the segmented regions. This indicates that the ESA plays a crucial role in generating initial prediction maps and positively influencing the quality of subsequent segmentation maps, demonstrating the effectiveness of ESA.

## Appendix D. Complexity Analysis

The computational complexity of the model is very important in polyp segmentation tasks, as obtaining segmentation information more quickly and accurately from real-time video streams would significantly enhance diagnostic and treatment efficiency for doctors performing colonoscopies on polyp tissues.

In this section, we compare PAAN with four state-of-the-art articles published after 2021. The comparison results are presented in Table 5, where "Macs" denotes cumulative operations or floating-point arithmetic operations. The results indicate that our network achieves superior performance compared to previous methods, despite its relatively lower computational and parameter requirements. This validates the effectiveness of the proposed network architecture in reducing computational complexity.

Table 5: Comparison of the computational complexity and parameter count of PAAN with other state-of-the-art methods.

| Model | UACANet-S | UACANet-L | CaraNet | FAPN | Ours |
|---|---|---|---|---|---|
| Macs(G) | 12.04 | 59.57 | 21.7 | 27.02 | 20.94 |
| Params(M) | 26.9 | 69.16 | 46.64 | 35.61 | 33.72 |

## Appendix E. Uncertainty Analysis

Deep neural networks often start training from random initial values, so even models trained with the same parameters and network may yield varying results. To further investigate the training effects of models under different random initial values and data sequences, this section conducted five training sessions with different random initial values and image orders under the same experimental conditions. The training results are shown in Table 6. From the table, it can be observed that PAAN achieved similar high-level results under different training conditions, with relatively small standard deviations. This demonstrates that the quality of network generation in this study is not derived from randomness but rather from reproducible experimental outcomes based on the network structure.

Table 6: Uncertainty analysis of our network trained with random initial values, where the mean is denoted by $\mu$ and the standard deviation is denoted by $\sigma$.

| Dataset | $\mu \pm \sigma$ | mDice ↑ | mDice ↑ | MAE ↓ | $F_\beta^w$ ↑ | $S_\alpha$ ↑ | $E_\phi^{mean}$↑ |
|---------|------------------|---------|---------|-------|---------------|--------------|------------------|
| Kvasir | $\mu$ | 0.9395 | 0.8955 | 0.0156 | 0.9330 | 0.9401 | 0.9758 |
|  | $\sigma$ | 0.0043 | 0.0031 | 0.0022 | 0.0046 | 0.0031 | 0.0038 |
| CVC-ClinicDB | $\mu$ | 0.9322 | 0.8819 | 0.0079 | 0.9297 | 0.9433 | 0.9818 |
|  | $\sigma$ | 0.0036 | 0.0044 | 0.0019 | 0.0041 | 0.0049 | 0.0041 |
| ETIS | $\mu$ | 0.8102 | 0.7311 | 0.0102 | 0.7860 | 0.8797 | 0.9344 |
|  | $\sigma$ | 0.0052 | 0.0077 | 0.0019 | 0.0072 | 0.0042 | 0.0051 |
| CVC-ColonDB | $\mu$ | 0.7837 | 0.7129 | 0.0336 | 0.7811 | 0.8522 | 0.8928 |
|  | $\sigma$ | 0.0047 | 0.0089 | 0.0009 | 0.0048 | 0.0030 | 0.0041 |
| CVC-300 | $\mu$ | 0.9236 | 0.8658 | 0.0044 | 0.9180 | 0.9454 | 0.9901 |
|  | $\sigma$ | 0.0044 | 0.0051 | 0.0004 | 0.0067 | 0.0038 | 0.0023 |

## Appendix F. More Ablation Study

There are more visualization results of the ablation experiments in Figure 6, which provide a direct comparison of the effectiveness of each module.

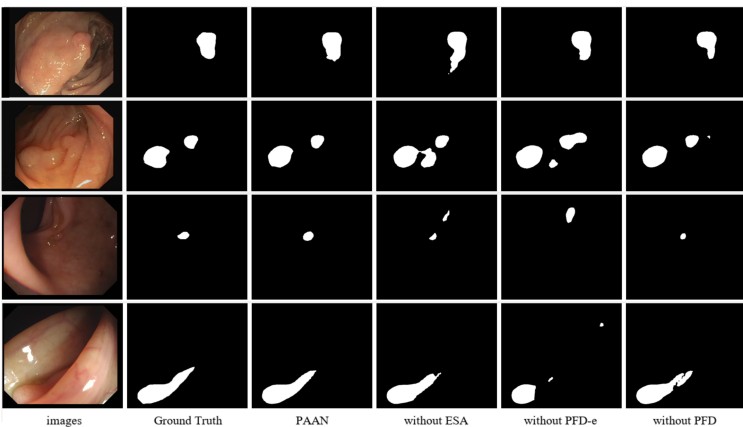

Figure 6: More visualization results of the ablation experiments.

