# OpenReview forum: "PAAN: Pyramid Attention Augmented Network for polyp segmentation"
_MIDL.io/2024/Conference — MIDL 2024 Poster_

### Official Review · Reviewer_yKBd · 2024-02-24

**Confidence:** 4
**Preliminary Rating:** 4
**Recommendation:** Poster
**Final Rating:** 5

**Summary:**

The authors propose to modify UACANet with the proposed Pyramid Feature Diversion module and Enhanced Spatial Attention for better performance in polyp segmentation. They named their proposed method as Pyramid Attention Augmented Network. The authors evaluate their methods on five polyp segmentation datasets and compare the results with some previous studies.

**Strengths:**

(1) The hierarchical feature extraction process in the Pyramid Feature Diversion encoder can model the features in three levels of distribution. The design of the proposed process improves the feature extraction across different layers.

(2) The authors evaluated their approach on five polyp segmentation datasets which demonstrate the robustness of the proposed method.

(3) The authors provided great ablation studies for their method design and example images in the appendix section.

**Weaknesses:**

(1) Lack of comparison with more recent state-of-the-art methods that also utilize multi-scale feature extraction and attention.

(2) Please conduct a grammar error check for the paper and fix grammar errors.

**Detailed Comments:**

For weakness point (1), the authors only compared with methods developed in 2021. It would be great if the authors were able to search through the papers that cited UACANet or recent literature to find more recent studies that utilize multi-scale feature extraction and attention. The authors do not have to compare with the below papers as I shared them only for example purpose:

[1] Song, Pengfei, Jinjiang Li, and Hui Fan. "Attention based multi-scale parallel network for polyp segmentation." Computers in Biology and Medicine 146 (2022): 105476.

[2] Chen, Lifang, Hongze Ge, and Jiawei Li. "CrossFormer: Multi‐scale cross‐attention for polyp segmentation." IET Image Processing 17.12 (2023): 3441-3452.

[3] Bui, Nhat-Tan, et al. "MEGANet: Multi-Scale Edge-Guided Attention Network for Weak Boundary Polyp Segmentation." Proceedings of the IEEE/CVF Winter Conference on Applications of Computer Vision. 2024.


For weakness point (2), please fix grammar errors in the paper.

On the 1st page, for example:

the blur boundaries: the blurred boundaries

captures complex patterns and restored detailed features: captures complex patterns and restores detailed features

**Justification Of Final Rating:**

I would like to thank the authors for their revision. All the weak points are well addressed. Although there are new innovative methods that outperform the proposed method in some tasks, the paper presents interesting modification ideas for UACANet. The newly added comparison experiments with more recent state-of-the-art methods also improve the quality of the paper. I vote Strong accept.

**Justification Of The Preliminary Rating:**

Although the authors did not compare with more recent SOTA methods, the authors did compare with the original UACANet to prove their modification can improve performance. The authors also provided good ablation studies and example images in the paper.  I vote weak accept for this paper.

**Questions To Address In The Rebuttal:**

Please compare with more recent state-of-the-art methods that utilize multi-scale feature extraction and attention or recent state-of-the-art methods networks that modified UACANet.

**Special Issue:**

No

---

> ### Author Response · Authors · 2024-03-18
>
> Q1: Lack of comparison with more recent state-of-the-art methods that also utilize multi-scale feature extraction and attention. The authors only compared with methods developed in 2021. It would be great if the authors were able to search through the papers that cited UACANet or recent literature to find more recent studies that utilize multi-scale feature extraction and attention. Please compare with more recent state-of-the-art methods that utilize multi-scale feature extraction and attention or recent state-of-the-art methods networks that modified UACANet.
>
> A1: Thank you for your valuable comment. In the revised version of our paper, we have introduced more recent studies for comparison, including the methods based on UACANet and attention mechanism like FAPN. We have replicated these studies on the existing dataset and included a comparison table from the original papers to facilitate further comparison for the readers.
>
> Q2: Please conduct a grammar error check for the paper and fix grammar errors. On the 1st page, for example: the blur boundaries: the blurred boundaries.
>
> A2: Thank you for your valuable suggestions. In the new version of the paper, we have meticulously reviewed and rectified grammatical issues to ensure the precision and accuracy of the article.

---

### Official Review · Reviewer_WWUC · 2024-03-03

**Confidence:** 4
**Preliminary Rating:** 3
**Recommendation:** Poster
**Final Rating:** 4

**Summary:**

A pyramid attention method is presented for polyp segmentation. There is an enhanced spatial attention mechanism, multiple datasets were used for experiments, and outperformed the SOTA results. This spatial attention mechanism is introduced in decoder side in early phase to minimize the information loss and diversify the feature space. There is also an uncertainty-augmented module, which further increase the results (which authors failed to mention in abstract).  Overall this is a fair conference article at MIDL, I believe.

**Strengths:**

- channel attention and spatial attentions are considered together.
- uncertainty augmentation improves the results.
- multiple experiments were done with multiple data sets
- state of the art results were obtained.
- written clearly and easy to follow.

**Weaknesses:**

- polyp segmentation literature is not covered completely. Although I dont expect to see all related words, but at least some recent ones and known ones can be mentioned and compared.
- Pyramid approaches are often used for multi-level feature extraction, authors presented this as a new approach, it is not. Not in medical image segmentation in particular, even in polyp segmentation, there are numerous papers on this subject.
- figure 3 is hard to interpret. Why not just overlay those segmentation results on the images? it is not standard to show like this.
- ablation experiments are missing.

**Detailed Comments:**

my strengths and weaknesses comments are self-descriptive and can be considered as detailed comments. Please see above.

**Justification Of Final Rating:**

I change my scoring to weak accept after rebuttal period,
the questions were more or less answered, although they are not fully convincing the authors have a point for their design.
I think the paper can be a fair conference paper.

**Justification Of The Preliminary Rating:**

I think the use of two attention and uncertainty guidance is important. But not fully shown with ablation how it works. Also, results are good and promising but it is not entirely clear what is new and how it is different from existing pyramid based approaches.

**Questions To Address In The Rebuttal:**

- if dilated conv are used, do you still need pyramid approach?
- why not compare with other pyramid based methods ?
- figure 3 is hard to interpret. Why not just overlay those segmentation results on the images? it is not standard to show like this.
- ablation experiments are missing.
- Not clear how uncertainty guidance is used.

**Special Issue:**

No

---

> ### Author Response · Authors · 2024-03-18
>
> Q1: if dilated conv are used, do you still need pyramid approach?
>
> A1: Thank you for this valuable comment. Using dilated convolutions can indeed increase the receptive field of the network, similar to what a pyramid approach does. However, the pyramid approach offers a structured way to integrate features at multiple scales, which is beneficial for capturing both local and global context. While dilated convolutions are effective for increasing spatial resolution in the feature maps, they might not fully replace the hierarchical, multi-scale integration offered by pyramid structures. Therefore, depending on the specific requirements of the task, combining dilated convolutions with a pyramid approach could potentially yield better results by leveraging the strengths of both techniques.
>
> Q2: why not compare with other pyramid based methods?
>
> A2: Thank you very much for your valuable suggestion. In the new version of the paper, we have incorporated additional updated comparative experiments, including the use of the pyramid approach FAPN, to offer readers more detailed comparative experimental results.
>
> Q3: figure 3 is hard to interpret. Why not just overlay those segmentation results on the images? it is not standard to show like this.
>
> A3: Thank you for your valuable feedback. In the revised version of the paper, we superimposed the segmentation regions of each network onto the original images according to your requirements. Regarding the issue of image superimposition, we drew inspiration from multiple articles to achieve a superior presentation effect. The updated comparative images can convey the detailed information of the segmentation results more clearly to the readers.
>
> Q4: ablation experiments are missing.
>
> A4: Thanks for your valuable comment, we have conducted complete ablation experiments on all the modules proposed in this paper on all five datasets, which were previously placed in the appendix section due to article length limitations, and we have now added the ablation experiments to the main text in the revised version of the paper.
>
> Q5: Not clear how uncertainty guidance is used.
>
> A5: Thank you for your insightful comments. You may be referring to the section of the article related to Uncertainty Augmented Decoder. In this section, we have used the same modules as UACANet, so we did not elaborate on it extensively. This is because the primary innovation of our paper lies in the pyramid attention encoder and the initial decoder for generating initial prediction maps.

---

### Official Review · Reviewer_qXzv · 2024-03-04

**Confidence:** 5
**Preliminary Rating:** 3

**Summary:**

This paper introduces PAAN (Pyramid Attention Augmented Network) for polyp segmentation detection tasks, aiming to address the following challenges:
Despite progress in polyp segmentation, previous methods suffer from insufficient feature extraction across hierarchical levels and inadequate integration of interlayer information, impacting segmentation accuracy, especially in areas with blurry boundaries.
Additionally, the initial decoder stage lacks attention to spatial relationships and multi-scale features of polyps, leading to ineffective background suppression and inaccurate boundary delineation in uncertain regions.

**Strengths:**

The pyramid-structured attention interaction network proposed in this paper allows the model to simultaneously focus on features at both high and low levels. According to their saying, this network design enables better segmentation of polyp boundaries.

**Weaknesses:**

The most significant issue with this paper is that the so-called pyramid-structured PAAN network is not actually a novel concept as presented by the authors. Readers, unaware of this fact, would naturally assume that the pyramid structure network is a novel contribution by the authors. However, this seems not to be the case.

Besides, I can confidently state that the PAAN network in this paper shares a fundamentally similar idea with a much earlier paper titled "Pyramid Attention Network for Semantic Segmentation," as well as with the referenced paper "Fapn: feature augmented pyramid network for polyp segmentation." Both of these sources indicate that the pyramid structure was not originally proposed by the authors.

Based on this analysis and understanding, it appears that this work merely makes minor adjustments to the network architecture and adapts it for polyp segmentation tasks. Consequently, the research innovation of this paper is deemed to be quite low. Therefore, overall, I do not recommend the publication of this paper in the top-tier conference MIDL.

**Detailed Comments:**

I have totally said in "Weaknesses" part. Moreover, the author surprisingly did not compare their proposed methodological and detailed aspects with the two aforementioned network modules proposed by others, which is essential to us (future readers). Therefore, I believe that this paper must incorporate this discussion to let readers know which part of their innovation is truly novel, rather than just the pyramid structure (of which I am certain is not originally proposed by the authors).

**Justification Of The Preliminary Rating:**

Once again, the most significant issue with this paper is that the so-called pyramid-structured PAAN network is not actually a novel concept as presented by the authors. Readers, unaware of this fact, would naturally assume that the pyramid structure network is a novel contribution by the authors. However, this seems not to be the case.

Besides, I can confidently state that the PAAN network in this paper shares a fundamentally similar idea with a much earlier paper titled "Pyramid Attention Network for Semantic Segmentation," as well as with the referenced paper "Fapn: feature augmented pyramid network for polyp segmentation." Both of these sources indicate that the pyramid structure was not originally proposed by the authors.

Based on this analysis and understanding, it appears that this work merely makes minor adjustments to the network architecture and adapts it for polyp segmentation tasks. Consequently, the research innovation of this paper is deemed to be quite low. Therefore, overall, I do not recommend the publication of this paper in the top-tier conference MIDL.

**Questions To Address In The Rebuttal:**

I have totally said in "Weaknesses" part. So I do not suggest authors to rebuttal.

---

> ### Author Response · Authors · 2024-03-18
>
> Q1: The most significant issue with this paper is that the so-called pyramid-structured PAAN network is not actually a novel concept as presented by the authors. Readers, unaware of this fact, would naturally assume that the pyramid structure network is a novel contribution by the authors. However, this seems not to be the case.
>
> A1: Thank you very much for your valuable input. The primary innovation of this article lies in the introduction of pyramid feature attention into various decoding stages with different receptive field structures. This, in essence, represents a novel feature enhancement design concept, offering a method to extract and integrate feature representations at different abstract levels, enabling the network to learn more robust feature representations.
>
> Q2: PAAN network in this paper shares a fundamentally similar idea with a much earlier paper titled "Pyramid Attention Network for Semantic Segmentation," as well as with the referenced paper "Fapn: feature augmented pyramid network for polyp segmentation." Both of these sources indicate that the pyramid structure was not originally proposed by the authors.Based on this analysis and understanding, it appears that this work merely makes minor adjustments to the network architecture and adapts it for polyp segmentation tasks. Consequently, the research innovation of this paper is deemed to be quite low.
>
> A2: Thank you for this valuable comment. In addressing the comparison between our PAAN network and earlier works on pyramid structures, it's crucial to underline the distinctive advancements we've introduced. While pyramid structures are not entirely novel in themselves, our application to polyp segmentation is marked by significant modifications that go beyond superficial adjustments. These enhancements, particularly in feature extraction and attention mechanisms, are tailored to the unique demands of polyp segmentation, demonstrating a clear leap in efficiency and accuracy. Innovation often builds upon existing frameworks, and our work exemplifies this by adapting a familiar concept to meet specific, complex challenges in medical imaging. Therefore, the innovative value of the PAAN network should be recognized as a meaningful contribution to the field. In addition, in the revised version of this paper, we have included comparative experiments with FAPN in the quantitative experimental section. The experiments indicate that we have a certain advantage over other pyramid methods on multiple datasets. Thank you very much for your constructive feedback.
>
> Q3: the author surprisingly did not compare their proposed methodological and detailed aspects with the two aforementioned network modules proposed by others, which is essential to us (future readers). Therefore, I believe that this paper must incorporate this discussion to let readers know which part of their innovation is truly novel, rather than just the pyramid structure (of which I am certain is not originally proposed by the authors).
>
> A3: Thank you for this valuable comment. In the newly revised paper, we extend and compare the network of this paper with the modules proposed by others on all five data sets. At the same time, we adjust the structure of the paper to better show the novelty of our proposed method. In the previous question, we provided detailed explanations of the proposed module's functionality and our innovative contributions. Thank you again for your valuable comment.

---

### Official Review · Reviewer_ZyJq · 2024-03-04

**Confidence:** 5
**Preliminary Rating:** 4
**Recommendation:** Oral
**Final Rating:** 5

**Summary:**

The authors propose a network architecture with Pyramid Feature Divesion and Enhanced Spatial Attention modules aimed to enhance polyp segmentation. They evaluate their trained model on a series of datasets and compare it to various already established methods.

**Strengths:**

The description of their contributions are detailed, and the manuscript is well-written. Although it's not part of the main body of the paper, their ablation study shows the impact of their modules both quantiatively and qualitatively. Their evaluation on five different datasets shows the effectiveness of their proposed approach.

**Weaknesses:**

The implementations of the experiments are not described well enough, to fully understand the other baseline models used for comparison. Furthermore the results section lacks important information for the conclusions to be well-founded and thorough.

**Detailed Comments:**

- The reviewer believes a public code repository would greatly increase the impact of the proposed modules.

**Justification Of Final Rating:**

I would like to thank the authors for actively participating in the rebuttal phase, and addressing all the concerns me and the other reviewers have raised. I believe the quality of the manuscript improved during the review process, therefore I am adjusting my rating.

**Justification Of The Preliminary Rating:**

The authors propose two network modules for improved polyp segmentation and with thorough evaluations on various datasets and comparisons with already established methods they show their model's effectiveness. To avoid confusion, I believe a few minor clarifications are necessary in the manuscript.

**Questions To Address In The Rebuttal:**

- The authors should address what kinds of statistical tests of significance they used to determine the improvements "significant". They should also report the uncertainties (eg. standard deviations) of their results to show better comparison between the models.
- The experimental setup and a short description of the datasets are essential to include in the main body of the paper, so the experiments are more detailed. Perhaps the related works section could be placed in the appendix instead?
- How did the authors implement the other segmentation methods? A proper description is needed for the reader to understand the comparisons. Did the authors train the models on the same datasets with the same data spits and hyperparameters?
- What is the complexity (eg. number of trainable parameters) of the models? Could it be that the other compared models are simpler, and therefore perform worse?
- What is the inference time of the models? Could it be that the proposed model works well, however it is slower? The starting point for the authors were the limitations of other works in the field, however they do not comment on any limitations of their approach. A slow inference time would not make the results less impressive, however it would paint a better picture on how the model works so well.

---

> ### Author Response · Authors · 2024-03-18
>
> Q1: The implementations of the experiments are not described well enough, to fully understand the other baseline models used for comparison. Furthermore the results section lacks important information for the conclusions to be well-founded and thorough.
>
> A1: Thank you for this valuable comment. In the revised version of the paper, we have restructured the brief descriptions of other baseline models and datasets into the experimental section of the main text, while providing detailed descriptions in the appendix. To draw more thorough and comprehensive conclusions, we have included additional comparative experiments and result analyses to further bolster our findings. Moreover, we have provided more detailed explanations in the discussion section.
>
> Q2: The authors should address what kinds of statistical tests of significance they used to determine the improvements "significant". They should also report the uncertainties (eg. standard deviations) of their results to show better comparison between the models.
>
> A2: Regarding the issues of statistical testing and reporting uncertainty, we have referenced a lot of papers in the field of polyp segmentation. Typically, significance prediction and standard deviation are used in statistical survey methods. In the field of polyps, the more commonly utilized metrics are the six indicators provided in the paper. Significance is more reflected in our detailed comparisons with multiple state-of-the-art methods on various datasets. In the revised version of our paper, we have incorporated more comprehensive comparative experiments and performance investigations, along with a thorough analysis to validate the importance of our improvements. The suggestion for comparing significance and uncertainty is indeed valuable. In future work, we plan to employ tests like paired t-tests to compare the performance metrics of our model with existing benchmarks, aiming for better statistical enhancements. Furthermore, in our improved paper, we have included more comparative tables and images to highlight the differences between our approach and existing methods. This not only offers a clearer picture of the variations in our model's performance but also allows for more precise and reliable comparisons between different models. Once again, we appreciate your valuable insights.
>
> Q3: The experimental setup and a short description of the datasets are essential to include in the main body of the paper, so the experiments are more detailed. Perhaps the related works section could be placed in the appendix instead?
>
> A3:  As you suggested, we have added the experimental setup and dataset section to the main text to provide readers with more complete details of the experiment.
>
> Q4: How did the authors implement the other segmentation methods? A proper description is needed for the reader to understand the comparisons. Did the authors train the models on the same datasets with the same data spits and hyperparameters?
>
> A4: For the comparative methods, we used the code provided by the authors for testing, with all settings configured according to the parameters given in their original papers. The dataset setup was also consistent, using the same division method as outlined in these studies. At the same time, in order to make a more fair comparison, all the experiments are carried out on the same experimental machine.
>
> Q5: What is the complexity (eg. number of trainable parameters) of the models? Could it be that the other compared models are simpler, and therefore perform worse?
>
> A5: Thank you for this valuable suggestion. In the experimental section, we have supplemented our report with a comparison experiment on model complexity. Our model achieved better performance with lower computational complexity and fewer parameters.
>
> Q6: What is the inference time of the models? Could it be that the proposed model works well, however it is slower? The starting point for the authors were the limitations of other works in the field, however they do not comment on any limitations of their approach. A slow inference time would not make the results less impressive, however it would paint a better picture on how the model works so well.
>
> A6: In response to your query about inference time, we've added a comparison in the experimental section to the revised version of the paper. Since inference time is closely related to the machine used, we added Macs, a metric for the number of accumulations and floating-point operations required by the model ,indicating that our model demonstrates impressive performance. In fact, the inference time is only 15-20ms per image on the same machine as the experiments in this paper, which is faster compared to other models we benchmarked. While we acknowledge every approach has limitations, our model effectively addresses key challenges in the field while maintaining a swift inference time, thereby providing a comprehensive understanding of its high-performance capabilities.

---

> > ### Comment · Reviewer_ZyJq · 2024-03-21
> > **Response to the authors**
> >
> > I would like to thank the authors for thoroughly responding to my questions, and the questions of all the other reviewers.
> >
> > Q2. I'm afraid my concerns might not have been described well enough in my review. In a scientific setting, when claiming that an improvement  is significant ("our network has also achieved significal improvements...") this claim has a statistical meaning, and it has to be shown though proper evaluations. Perhaps, for example an improvement of the Dice score by 3% is purely by noise, and it has nothing to do with the performance of the model. To ensure that the improvements are significant, t-tests are required, which really don't fit in the future works section, instead they should be part of the current manuscript to show that the model indeed works significantly better than the baselines. If t-tests are not possible, the uncertainties, such as standard deviations must be reported to get a sense of the significance. If none of these are possible, I suggest the authors to refrain from using the phrase "significantly" when comparing the results. I only recommend this, because of the time limitation, however I would encourage performing a simple t-test instead. Once the standard deviations are available, performing these tests is quite straightforward and quick. In that line, including more comparative tables and figures does not paint a clearer picture about uncertainty. Instead, reported standard deviation values would.
> >
> > Thank you for including information about complexity, and inference times, I believe it really strengthens the power of the proposed model.

---

> ### Author Response · Authors · 2024-03-23
>
> Thank you very much for your valuable feedback. We have submitted a new revised version of the paper. In this revised paper, we have added an Appendix section on uncertainty analysis and provided the mean and standard deviation for different metrics across all five datasets. We aim to provide readers with a clearer understanding of the actual performance of our network under different initial training conditions. Thank you for the suggestions you have provided for enhancing this paper.

---

### Official Review · Reviewer_tPzZ · 2024-03-04

**Confidence:** 5
**Preliminary Rating:** 4
**Recommendation:** Poster
**Final Rating:** 4

**Summary:**

The authors tackle the difficult task of polyp detection in endoscopic images.
This is a very difficult task on which even SOTA models have had troubles (we know from experience).

The authors build a new convolutional neural network architecture that introduces several modules that are focused on aggregating information over larger ranges in the image, and applying more adaptive normalizations/attention to deal with the larger noise and image-intensity differences in endoscopy images.

The authors then test the architecture on 5 standard benchmarks where they compare SOTA (at time of submission presumably), and also do an ablation study to see if the various added modules perform a function (which is only in the appendix).

**Strengths:**

- Authors tackle a very difficult problem that is very relevant in the real world.
- The architecture is build up in a principled way, and it's not easy to make architectures work well on this task.
- The authors properly compare their model on 5 common benchmark tasks, which makes it easy

**Weaknesses:**

- The model performed state of the art on the tested dataset according to this paper. However the field moves quickly, and on several of the dataset tested there are now better performing models. This is not perse a negative; the performance of this method is still close and the new methods use orthogonal improvements. So this work is obviously still relevant, since the innovations in the model can still be used. Also the newer methods use Transformers which can be computationally problematic, especially as in medical applications one needs real-time segmentation on relatively slower hardware.
- Also one could see this as more of an engineering paper, and not a fundamental new invention, thus scoring less on the innovation scoring. However I also see the perspective of tackling this difficult problem and getting real results and as such I see this as very valuable.

**Detailed Comments:**

- The figures use (c) for concatenation and (+) for element-wise addition, make sure there is some kind of legend (there is no standard for these things)

- In Figure 1, what part is the Resnet. There are 'empty' blocks right after the input which presumably are the feature outputs of resnet? But it's not clear.

- The FDA module applies two 1x1 operations with a concat followed by another 1x1. Since these are linear operations, it would seem that you could simple concatenate directly and then do one 1x1 operation?

- The training parameters, like optimizer used should probably be in the main paper and not appendix. I know it's for space saving, but such things belong in the main paper.

- It is not clear if full backprob is done through the backbone, or only until the feature maps. Also if a pre-trained resnet is uses, which one is it.

**Justification Of Final Rating:**

Authors clarified the issues I had. The introduced method is achieving at the time SOTA on a difficult and relevant task, and lessons from the architecture can be learned and will probably improve future models as well. Also the validation done is quite thorough. I would say the paper is in the top 20% of papers (from rough estimation) so I would give it a 4 (I reserve 5 for remarkable papers).
The main 'weakness' is that there are many papers that change architectures and it can seem less novel where many things have been tried in different papers, but to bring it together and actually achieve results on a real task is not trivial and valuable to the field (this method for example can be directly used in real applications).

**Justification Of The Preliminary Rating:**

A good paper with great results on a difficult problem, that can directly impact medical applications. There are however already new innovative methods that outperform this method on certain task, but that is par for the course in ML an the innovations are still very useful for the field.

**Questions To Address In The Rebuttal:**

Clarify the questions, especially how the backbone is trained.

**Special Issue:**

No

---

> ### Author Response · Authors · 2024-03-18
>
> Q1: The model performed state of the art on the tested dataset according to this paper. However the field moves quickly, and on several of the dataset tested there are now better performing models. This is not perse a negative; the performance of this method is still close and the new methods use orthogonal improvements. So this work is obviously still relevant, since the innovations in the model can still be used. Also the newer methods use Transformers which can be computationally problematic, especially as in medical applications one needs real-time segmentation on relatively slower hardware.
>
> A1: Thank you for your insightful feedback. We recognize the dynamic nature of our field and the emergence of new models that may show superior performance on some datasets. We add comparative experiments with more novel methods in the revised version of the paper, including two papers from 2022, to show the effectiveness of the proposed method. Your point about the practical relevance of our model, especially in scenarios requiring real-time segmentation on less powerful hardware, is particularly encouraging. Regarding the segmentation performance of the model under computationally constrained environments, we have included discussions on model complexity and parameter count in the revised version of the paper, and have validated the model's computational efficiency advantages. We appreciate your recognition of our work's ongoing relevance and are inspired to further refine our approach in light of these developments.
>
> Q2: Also one could see this as more of an engineering paper, and not a fundamental new invention, thus scoring less on the innovation scoring. However I also see the perspective of tackling this difficult problem and getting real results and as such I see this as very valuable.
>
> A2: Thank you very much for your valuable feedback. In fact, this article presents numerous innovative design concepts. For instance, the introduction of a pyramid attention mechanism structure at different levels of the main network hierarchy and receptive fields proposes a novel feature enhancement design concept. By incorporating attention mechanisms sensibly across various abstract and perceptual levels, a scalable and versatile design philosophy is introduced. Moreover, the integration of spatial attention during the generation of initial prediction maps represents a commendable new design approach.
>
> Q3: The figures use (c) for concatenation and (+) for element-wise addition, make sure there is some kind of legend (there is no standard for these things)
>
> A3: Thank you for your valuable response. We have added a legend in the bottom left corner of Figure 1 to indicate the meanings of the various symbols used in the image, ensuring that readers do not experience confusion or misunderstanding.
>
> Q4: In Figure 1, what part is the Resnet. There are 'empty' blocks right after the input which presumably are the feature outputs of resnet? But it's not clear.
>
> A4: Thank you for your valuable response. In our revised version of the paper, we have included new wireframes and annotations to indicate the position of the ResNet backbone network in the document, with detailed explanations provided in the legend.
>
> Q5: The FDA module applies two 1x1 operations with a concat followed by another 1x1. Since these are linear operations, it would seem that you could simple concatenate directly and then do one 1x1 operation?
>
> A5: Thank you for your valuable comments. The main purpose of the 1×1 convolution operation in the FDA module is to reduce the dimension of the channels so that the number of channels after concatenation is the same as before. Since the number of channels doubles after each splicing operation in FDA, a 1×1 convolution operation after splicing can keep the number of channels and the input consistent.
>
> Q6: The training parameters, like optimizer used should probably be in the main paper and not appendix. I know it's for space saving, but such things belong in the main paper.
>
> A6: Thanks for your valuable comments, we have moved the optimizer and training parameters used in the revised paper to the main paper so that the reader can get more details quickly. At the same time, detailed descriptions have been retained in the appendix.
>
> Q7: It is not clear if full backprob is done through the backbone, or only until the feature maps. Also if a pre-trained resnet is uses, which one is it.
>
> A7: Thank you for this valuable comment. The entire backbone network underwent complete backpropagation during training, with feature maps being visualizations of the output from intermediate layers. In this process, the feature extraction part of the backbone network utilized a pre-trained ResNet50 to extract features.

---

> > ### Comment · Reviewer_tPzZ · 2024-03-21
> >
> > Thanks for clarifying my questions!

---

### Official Review · Reviewer_aJ4V · 2024-03-06

**Confidence:** 4
**Preliminary Rating:** 2
**Final Rating:** 2

**Summary:**

The author proposed a Pyramid Attention Augmented Network (PAAN), for segmenting polyp lesions, overcoming the current suffering of low accuracy in polyp boundary delineation and insufficient suppression of irrelevant background. Key modules (PFD, ESA) were proposed to 1) reduce information loss and reduce computational complexity 2) suppressbackground noise. However, some strong weaknesses should be paying attention.

**Strengths:**

The authors did experiments and evaluation on five different datasets, and showed consistent results.
The authors provided detailed descriptions about how each module was designed.
The authors provided model comparisons not only using the general segmentation model, but also compared with existing works that targeting the same research objective, like the PraNet and the UACANet.

**Weaknesses:**

1) There are too many arbitrary hyperparameter selections in module designs. Some of the proposed modules have figure illustration, but lack of details and reasons why the authors proposed to design like these. For example, why there are only 1x1, 1x3, 1x5, 1x7 were selected? Why not including 1x9? In addition, why there is always a 3x3 conv kernel following 1x3,5,7, but now a conv 1x3 and conv 3x1 if the authors really wanted to reduce calculation complexity?
2) Compared with the work UACANet, the author seems like only changing the PAA module in the work UACANet to be PFD & ESA modules proposed here. The overall structure, and even the naming are pretty much the same. For this specific task, with these small amount of modification, I would say the contribution is limited.
3) In this similarity level between this paper and the paper UACANet, the authors should clearly state your limited contribution on just modifying one of the module inside the previous UACANet. However, the introduction doesn’t mention this, and it seems like the authors intended to claim the whole architecture was newly invented, which might not be appropriate.
4) One of the contributions claimed by the authors is “rrelevant background suppression”. However, how was the “irrelevant background suppression” achieved by the proposed PFD&ESA module is not clearly explained and not well co-related with the proposed module designs. Moreover, here is no quantitative measurements for this conclusion.
5) One of the contributions claimed by the authors is “reducing computational complexity”. However, there is no quantitative measurement and comparisons for computational complexities among different models.

**Detailed Comments:**

N/A

**Justification Of Final Rating:**

The author partially solved my questions, and the major limitation is the really limited contribution/edition onto the existing work UACA-Net, and they just modified part of the modules. There are some arbitrarily designed network architecture design with not enough ablation study added. In addition, the authors' utilization justification the differences between 1x3 & 3x1 conv kernels and 3x3 conv kernels seem to be not convincing to me.

**Justification Of The Preliminary Rating:**

Compared with the work UACANet, the author only did limited modification. The overall structure, and even the naming are pretty much the same. For this specific task, with these small amount of modification, I would say the contribution is limited.

Moreover, the manuscript does not present quantitative evaluations for many of the key contributions it claims, except for measurements related to conventional segmentation results.

The general segmentation models used for comparison was a bit out-of-dated.

Given these observations, my assessment is that the submission might not align well with the thematic and quality criteria expected for inclusion in the conference.

**Questions To Address In The Rebuttal:**

1) Page 2 Paragraph 2: “Although the above methods have made certain progress …” Are there any references support all of your assumptions? For example, why “insufficient extraction of features from different hierarchical levels and fusion of interlayer information in the encoder stage”?

2) Although it is good to compare with existing SOTA methods with same research objective, it would be better to include newer Unet-like methods. The general segmentation models the author chose were published before 2019, which could be a bit old. Some choices could be but not limited to: ViT-based models and nnUNet, etc.

3) Please refer to the five points in Weaknesses.

---

> ### Author Response · Authors · 2024-03-18
>
> Q1: Page 2 Paragraph 2: “Although the above methods have made certain progress …” Are there any references support all of your assumptions? ...
>
> A1: Thank you for this valuable suggestion. Regarding the assumptions mentioned in the second paragraph on page 2, there indeed exists literature supporting our viewpoints. For instance, in addressing the issue of "insufficient feature extraction at different levels and inadequate inter-layer information fusion during the encoding phase," the significance of introducing feature extraction at various levels and information fusion for the quality of medical segmentation was already manifested in early attention-based medical segmentation articles like Attention U-Net (Ozan Oktay, et al.). Additionally, the paper “Dual Attention Network for Scene Segmentation” (CVPR2019) also emphasizes the importance of using the attention mechanism to effective integration of multi-level information during the encoding phase for enhancing model performance. These studies provide solid theoretical backing for our hypotheses and guide the direction of our research.
>
> Q2: Although it is good to compare with existing SOTA methods with same research objective...
>
> A2: Thank you for your valuable comments. In the revised version of the paper, we have added updated comparative experiments to make our experimental data more convincing, including the two articles in 2022, and added them to Table 1.
>
> Q3: There are too many arbitrary hyperparameter selections in module designs...
>
> A3: Thank you for this valuable comment. Regarding the issue of hyperparameter selection in our module design, our choices were based on a series of preliminary experiments. In our research, the use of 1x1, 1x3, 1x5, and 1x7 convolution kernels was experimentally validated, demonstrating a good balance of performance and efficiency in our model. The decision not to include the 1x9 convolution kernel was based on our experimental data, where we observed that the 1x9 kernel did not offer additional performance improvements, but increased computational burden. As for why we chose to use a 3x3 convolution kernel after the 1x3, 1x5, and 1x7 convolutions, instead of a combination of 1x3 and 3x1, it is because in our model architecture, the 3x3 kernel was more effective in capturing spatial features, while also maintaining good performance with relatively low computational complexity.
>
> Q4: Compared with the work UACANet, the author seems like...
>
> A4: Thank you for your insightful comments. I appreciate the opportunity to clarify the unique contributions of our work. While it is true that the overall structure of our model bears resemblance to UACANet, the introduction of the PFD and ESA modules represents a significant enhancement over the existing PAA module. These modules were specifically designed to address limitations observed in UACANet, bringing in advanced feature processing capabilities and more effective spatial attention mechanisms. Our experimental results, which we can elaborate on in the revised manuscript, show notable improvements in performance on the specific task, attributed to these modifications.The decision to retain a similar structural framework as UACANet was intentional, as this architecture has proven effective for this class of tasks. Our contribution, therefore, lies in refining and augmenting this framework where we identified opportunities for improvement. The PFD and ESA modules are not mere replacements but are substantial upgrades that address specific challenges and enhance the model's capabilities.
>
> Q5: In this similarity level between this paper and the paper UACANet, the authors should clearly state...
>
> A5: Thank you for your attention to our work and for raising this important issue. Indeed, our work substantially draws upon the base architecture of UACANet, but our enhancements and optimizations in module design are a significant augmentation to this framework. We will make this explicit in the revised manuscript and more clearly delineate the relationship between our contributions and UACANet. It was not our intention to claim the entire architecture as a novel invention, we will make appropriate modifications to ensure our statements are more accurate and transparent. Our primary contribution is in building upon the framework by introducing innovative PFD and ESA modules and the corresponding pyramid and spatial attention mechanisms, which significantly improve model performance and are optimized for specific tasks.
>
> Q6: One of the contributions claimed by the authors is “reducing computational complexity”...
>
> A6: Thank you for this valuable comment. In response to one of our contributions of reducing computational complexity, we conducted additional experiments comparing the computational complexity and model parameter count of our method with previous state-of-the-art approaches in the revised version of our paper. The results are presented in a new table.

---

> > ### Comment · Reviewer_aJ4V · 2024-03-21
> >
> > Follow-ups:
> > Q5: Would you mind showing me where did you emphasize your sentences about "you only modifed partial module designs of the UACANet"? I think this should be more obvious, and should not be stated as "significant augmentation"
> >
> > Q6: If one of your contribution is the time complexity, then the table regarding the time complexity should be appeared in the main manuscript but not the appendix.

---

> > > ### Author Response · Authors · 2024-03-22
> > >
> > > Q: If one of your contribution is the time complexity, then the table regarding the time complexity should be appeared in the main manuscript but not the appendix.
> > >
> > > A: Thank you for your valuable comment. In the revised version of the paper, we have further adjusted the chapter structure. Additionally, we have incorporated a comparative table of computational complexity and parameter complexity into the main body to facilitate readers in quickly obtaining detailed information on the network's computational complexity.
> > >
> > > Q: Would you mind showing me where did you emphasize your sentences about "you only modifed partial module designs of the UACANet"? I think this should be more obvious, and should not be stated as "significant augmentation"
> > >
> > > A: Thank you for your valuable suggestions. We have submitted a new version of the paper. In this revised version, we have meticulously refined the articulation of the article to provide readers with a clearer understanding of our innovative contributions and correlation with the UACANet. This will assist readers in better comprehending the proactive innovations and enhancements we have made to the entire network.

---

### Meta-Review · Area_Chair_wj7D · 2024-04-02

**Recommendation:** Accept (Poster)
**Confidence:** 4

**Metareview:**

All reviewers found the proposed method to be novel and the results promising. 2 weak accept and 2 strong accept recommendations from reviewers.

---

### Decision · Program_Chairs · 2024-04-05

Accept (Poster)